# Atmospheric bending effects in GNSS tomography

Gregor Möller[1] and Daniel Landskron[1]

[1]Department of Geodesy and Geoinformation, Vienna University of Technology, Vienna, Austria

**Correspondence:** Gregor Möller (gregor.moeller@tuwien.ac.at)

**Abstract.** In GNSS tomography, precise information about the tropospheric water vapor distribution is derived from integral measurements like ground-based GNSS slant wet delays (SWDs). Therefore, the functional relation between observations and unknowns, i.e. the signal paths through the atmosphere have to be accurately known for each station-satellite pair involved. For GNSS signals observed above 15 degrees elevation angle, the signal path is well approximated by a straight line. However, since electromagnetic waves are prone to atmospheric bending effects, this assumption is not sufficient anymore for lower elevation angles. Thus, in the following, a mixed 2D piecewise linear ray-tracing approach is introduced and possible error sources in reconstruction of the bended signal paths are analyzed in more detail. Especially, if low elevation observations are considered, unmodeled bending effects can introduce a systematic error of up to $10 - 20 ppm$, on average of $1 - 2 ppm$ into the tomography solution. Thereby, not only the ray-tracing method but also the quality of the a priori field can have a significant impact on the reconstructed signal paths, if not reduced by iterative processing. In order to keep the processing time within acceptable limits, a bending model is applied for the upper part of the neutral atmosphere. It helps to reduce the number of processing steps by up to $85\%$ without significant degradation in accuracy. Therewith, the developed mixed ray-tracing approach allows not only for a correct treatment of low elevation observations but is also fast and applicable for near real-time applications.

## 1 Introduction

For conversion of precise integral measurements into two- or three-dimensional structures, a technique called tomography has been invented. In the field of GNSS meteorology, the principle of tomography became applicable with the increasing number of Global Navigation Satellite System (GNSS) satellites and the build-up of densified ground-based GNSS networks in the 1990s (Raymond et al., 1994; Flores Jimenez, 1999). Since then, a variety of tomography approaches based on raw GNSS phase measurements (Nilsson, 2005), double difference residuals (Kruse, 2001), slant delays (Flores Jimenez, 1999; Hirahara, 2000) or slant integrated water vapor (Champollion et al., 2005) has been developed for the accurate reconstruction of the water vapor distribution in the lower atmosphere. An overview about the major developments within this field of research since Flores Jimenez (1999) is provided by Manning (2013).

While in most tomography approaches, observations gathered at low elevation angles are discarded (Bender et al., 2011; Champollion et al., 2005; Hirahara, 2000), straight line signal path reconstruction is sufficient for the determination of the path lengths. However, Bender and Raabe (2007) showed that especially low elevation observations can be a very useful source of information in GNSS tomography. Besides their information content about the lower troposphere, the additional observations

strengthen the observation geometry and therewith, contribute to a more reliable tomography solution. However, a correct treatment of low elevation observations requires more advanced ray-tracing algorithms. The first paper which deals with bended ray path reconstruction in GNSS tomography was published by Zus et al. (2015), with main focus on the reconstruction of the signal paths for delay estimation but also for assimilation of GNSS slant delays into numerical weather prediction systems.

Most recently, Aghajany and Amerian (2017) published their results about 3D ray-tracing in water vapor tomography and briefly analyzed its impact on the tomography solution.

Based on the existing studies, in the following, a more detailed discussion of possible error sources in signal path reconstruction is provided. Therefore, Sect. 2 describes the effect of atmospheric bending and its handling in GNSS signal processing. Section 3 describes the principles of GNSS tomography and how the basic equation of tomography is solved for wet refrac-

tivity. Section 4 introduces the concept for reconstruction of signal paths using ray-tracing techniques. Hereby, the modified piecewise linear ray-tracing approach is described - including its ability for reconstruction of the GNSS signal geometry. In Sect. 5, the defined ray-tracing approach is applied to real SWDs and its impact on the tomography solution is assessed and validated against radiosonde data. Section 6 concludes the major findings.

## 2    Atmospheric bending effects in GNSS signal processing

The effect of atmospheric bending on GNSS signals is related to the propagation properties of electromagnetic waves. In vacuum, GNSS signals travel with the velocity of light. When entering into the atmosphere, the electromagnetic wave velocity changes, dependent on the electric permittivity ($\epsilon$) and magnetic permeability ($\mu$) of the atmospheric constituents and the frequency of the electromagnetic wave. The ratio between the velocity of light $c$ in vacuum and the velocity $\nu$ in a medium defines the refractive index $n$.

$$n = \frac{c}{\nu} = \sqrt{\frac{\epsilon \cdot \mu}{\epsilon_0 \cdot \mu_0}} \tag{1}$$

For signals in the microwave frequency-band, $n$ ranges from 0.9996 to 1.0004. Thus, $n$ is usually replaced by refractivity $N$, expressed in mm/km (ppm).

$$N = 10^6 \cdot (n - 1) \tag{2}$$

The GNSS signal delay in the lower atmosphere, also known as slant total delay ($STD$), is related to refractivity by the

following equation (Bevis et al., 1992):

$$STD = 10^{-6} \cdot \int_R N \cdot ds + \left[ \int_R ds - \int_S ds \right] \tag{3}$$

The first term of Eq. (3) describes the change in travel time due to velocity changes along the true ray path R. The second term (about three orders of magnitude smaller than the first term) is related to the difference in geometrical path length between the true (R) and the chord signal path (S). According to Dalton's law, the refractivity of air can be split up into a hydrostatic and a

wet component: $N = N_h + N_w$. Therewith, the GNSS signal delay reads:

$$STD = SHD + SWD = 10^{-6} \cdot \int_R N_h \cdot ds + 10^{-6} \cdot \int_R N_w \cdot ds + \left[ \int_R ds - \int_S ds \right]. \quad (4)$$

The slant wet delay $(SWD)$ depends on the wet refractivity along the true ray path $R$.

$$SWD = 10^{-6} \cdot \int_R N_w \cdot ds \quad (5)$$

The slant hydrostatic delay $(SHD)$ results from the the hydrostatic refractivity along $R$ and, by definition, from the additional path length due to atmospheric bending.

$$SHD = 10^{-6} \cdot \int_R N_h \cdot ds + \left[ \int_R ds - \int_S ds \right] \quad (6)$$

While signal path $S$ follows from the straight line geometry between satellite and receiver, the true signal path $R$ depends in addition on the hydrostatic and the wet refractivity distribution along the signal path (see Sect. 3 for more details).

In GNSS signal processing, the integral along the signal path is usually replaced by the zenith delay and a mapping function. Therefore, Eq. (4) is rewritten as follows:

$$STD(\varepsilon, \alpha) = SHD + SWD = ZHD \cdot mf_h(\varepsilon) + ZWD \cdot mf_w(\varepsilon) + G(\varepsilon, \alpha) \quad (7)$$

where $ZHD$ is the zenith hydrostatic delay, $ZWD$ is the zenith wet delay and $mf_h$ and $mf_w$ are the corresponding mapping functions, which describe the elevation ($\varepsilon$) dependency of the signal delay. The elevation and azimuth ($\alpha$) dependent first-order

horizontally asymmetric term $G(\varepsilon, \alpha)$ reflects local variations in the atmospheric conditions: see MacMillan (1995), Chen and Herring (1997) or Landskron and Böhm (2018). In practice, e.g. when using VMF1 mapping function (Böhm et al., 2006) or similar mapping concepts, the tropospheric delay due to atmospheric bending is absorbed by the hydrostatic mapping function term $mf_h$. Comparisons between ray-traced $SHD(\varepsilon)$ and 'mapped' $SHD(\varepsilon) = ZHD \cdot mf_h(\varepsilon)$ slant hydrostatic delays reveal that about 97 % of the atmospheric bending effect is compensated by the VMF1 hydrostatic mapping function (see Appx. A

for further details).

## 3   The principles of GNSS tomography

According to Iyer and Hirahara (1993), the general principle of tomography is described as follows:

$$f_s = \int_R g_s \cdot ds \quad (8)$$

where $f_s$ is the projection function, $g_s$ is the object property function and $ds$ is a small element of the ray path $R$ along which

the integration takes place. In GNSS tomography, $g_s$ is usually replaced by wet refractivity $N_w$, and integral measure $f_s$ by $SWD$ (the prefactor of $10^6$ vanishes if $ds$ is provided in kilometer and $SWD$ in millimeter).

$$SWD = \int_R N_w \cdot ds \quad (9)$$

A full non-linear solution of Eq. (9) for wet refractivity is not of practical relevance since according to Fermat´s principle, first order changes of the ray path lead to second order changes in travel time. In consequence, by ignoring the path dependency in the inversion of $N_w$ along $ds$ and by assuming the ray path as a straight line, a linear tomography approach can be defined which is well applicable to $SWDs$ above 15 degrees elevation angle (Möller, 2017). However, with decreasing elevation angle, the true signal path deviates significantly from a straight line. In consequence, by ignoring atmospheric bending, a systematic error is introduced in the tomography solution. In order to overcome this limitation, in the following an *iterative tomography approach* is defined in which the bended signal path is approximated by small line segments. Similar to the linear tomography approach, thereby the neutral atmosphere or parts of it are discretized in volume elements (voxels) in which the refractivity $N_{w,k}$ in each voxel $k$ is assumed as constant. Consequently, Eq. (9) can be replaced by:

$$SWD = \sum_{k=1}^{m} N_{w,k} \cdot d_k \tag{10}$$

where $d_k$ is the travelled distance in each voxel. Assuming $l$ observations and $m$ voxels, a linear equation system can be set up. In matrix notation it reads:

$$\boldsymbol{SWD} = \boldsymbol{A} \cdot \boldsymbol{N_w} \tag{11}$$

where $\boldsymbol{SWD}$ is the observation vector of size $(l,1)$, $\boldsymbol{N_w}$ is the vector of unknowns of size $(m,1)$ and $\boldsymbol{A}$ is a matrix of size $(l,m)$ which contains the partial derivatives of the slant wet delays with respect to the unknowns, i.e. the travelled distances $d_k$ in each voxel.

$$\boldsymbol{A} = \begin{bmatrix} \frac{\delta SWD_1}{\delta N_{w,1}} & \cdots & \frac{\delta SWD_1}{\delta N_{w,m}} \\ \vdots & \ddots & \vdots \\ \frac{\delta SWD_l}{\delta N_{w,1}} & \cdots & \frac{\delta SWD_l}{\delta N_{w,m}} \end{bmatrix} \tag{12}$$

Solving Eq. (11) for $\boldsymbol{N_w}$ requires the inversion of matrix $\boldsymbol{A}$.

$$\boldsymbol{N_w} = \boldsymbol{A}^{-1} \cdot \boldsymbol{SWD} \tag{13}$$

The inverse $\boldsymbol{A}^{-1}$ exists if $\boldsymbol{A}$ is squared and if the determinant of $\boldsymbol{A}$ is non-zero, otherwise matrix $\boldsymbol{A}$ is called singular. Unfortunately, singularity appears in GNSS tomography in most cases since the observation data is 'incomplete' and matrix $\boldsymbol{A}$ is not of full rank. Therewith, Eq. (13) becomes ill-posed, i.e. not uniquely solvable. In order to find a solution which preserves most properties of an inverse, in the following matrix $\boldsymbol{A}$ is replaced by pseudo inverse $\boldsymbol{A}^+$. According to Hansen (2000) the pseudo inverse is defined as follows:

$$\boldsymbol{A}^+ = \boldsymbol{V} \cdot \boldsymbol{S}^{-1} \cdot \boldsymbol{U^T} \tag{14}$$

where U and V are orthogonal, normalized left and right singular vectors of A and matrix S is a diagonal matrix, which contains the singular values in descending order. In case a priori information $(\boldsymbol{N_{w0}})$ can be made available, it enters the tomography solution as first guess as follows:

$$\boldsymbol{N_w} = \boldsymbol{N_{w0}} + \boldsymbol{V} \cdot \boldsymbol{S}^{-1} \cdot \boldsymbol{U^T} \cdot \boldsymbol{A^T} \cdot \boldsymbol{P} \cdot (\boldsymbol{SWD} - \boldsymbol{A} \cdot \boldsymbol{N_{w0}}) \tag{15}$$

where matrix $U$, $V$ and $S$ are obtained by singular value decomposition of matrix $A^T \cdot P \cdot A + P_c$. The weighting matrices $P$ and $P_c$ are defined as the inverse of the variance-covariance matrix $C$ for the observations and $C_c$ for the first guess, respectively. Assuming, that the observations are uncorrelated, the non-diagonal elements of $C$ and $C_c$ are zero and the diagonal elements are defined as follows:

$$\sigma_C^2 = \sin^2 \varepsilon \cdot \sigma_{ZWD}^2 \tag{16}$$

$$\sigma_{C_c}^2 = \left( \frac{\partial N_w}{\partial T} \cdot \sigma_T \right)^2 + \left( \frac{\partial N_w}{\partial q} \cdot \sigma_q \right)^2 + \left( \frac{\partial N_w}{\partial p} \cdot \sigma_p \right)^2 \tag{17}$$

whereby $\sigma_{ZWD} = 2.5mm$ reflects the uncertainty of the $ZWD$. The values for $\sigma_T$, $\sigma_q$ and $\sigma_p$ were taken from height-dependent error curves for pressure $(p)$, temperature $(T)$ and specific humidity $(q)$ as provided by Steiner et al. (2006) for the ECMWF (European Centre for Medium-Range Weather Forecasts) analysis data. For further details, the reader is referred to Möller (2017).

## 4    Reconstruction of GNSS signal paths

Assuming that the geometrical optics approximation is valid and that the atmospheric conditions change only inappreciably within one wavelength, the signal path is well reconstructible by means of ray-tracing shooting techniques (Hofmeister, 2016; Nievinski, 2009). Thereby, the basic equation for ray-tracing, the so-called Eikonal equation, has to be solved for obtaining optical path length $L$.

$$||\nabla L||^2 = n(\boldsymbol{r})^2 \tag{18}$$

From Eq. (18), a number of 3D and 2D ray-tracing approaches have been derived for the reconstruction of ground-based and space-based GNSS measurements and of their signal paths through the atmosphere (Hobiger et al., 2008; Zou et al., 1999).

The main difference between both observation types is related to the observation geometry. While for space-based GNSS observations derived in limb sounding, the bending angle is usually described as function of impact parameter $a$, for ground-based observations elevation and azimuth angle are used for characterizing the signal geometry. In consequence, the optimal ray-tracing approach will be significantly different for various observation geometries.

In order to find an optimal approach for operational analysis of ground-based measurements, Hofmeister (2016) carried out a number of exploratory comparisons. Based on the outcome, the 2D piecewise linear ray-tracer was defined as the optimal reconstruction tool for the iterative reconstruction of the atmospheric signal delays including atmospheric bending. It is limited to positive elevation angles but it is fast and almost as accurate as the 3D ray-tracer. However, for the use in GNSS tomography, the ray-tracing approach had to be further modified. In the following, the developed ray-tracing approach but also its impact on the GNSS tomography solution are discussed in more detail.

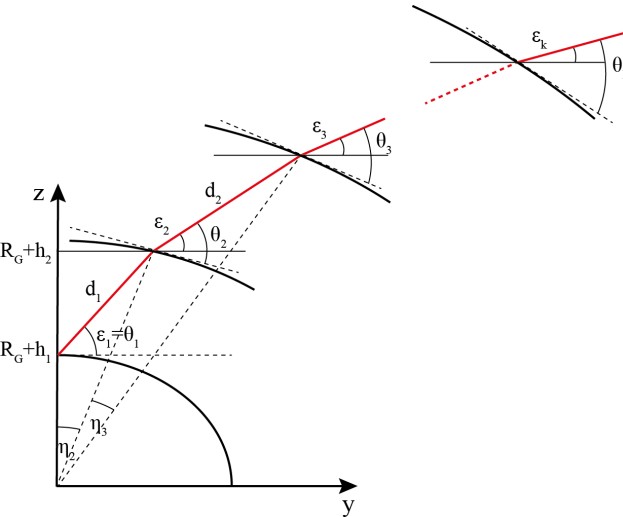

**Figure 1.** Geometry of the ray-tracing approach with the geocentric coordinates (y, z), the geocentric angles $(\eta, \theta)$, elevation angle $\varepsilon$ and $d$ as the distance between two consecutive ray points

### 4.1 Piecewise linear ray-tracer

The starting point for the 2D piecewise linear ray-tracer is the receiver position in ellipsoidal coordinates $(\varphi_1, \lambda_1, h_1)$, the 'vacuum' elevation angle $\varepsilon_k$ (see Fig. 1) and the azimuth angle $\alpha$ under which the satellite is observed. In case of GNSS tomography, these parameters can be determined with sufficient accuracy from satellite ephemerides and the receiver position - assuming straight line geometry. Therewith, the initial parameters for ray-tracing (see Fig. 1), i.e. the geocentric coordinates $(y_1, z_1)$ and the corresponding geocentric angles $(\eta_1, \theta_1)$ read:

$$y_1 = 0 \tag{19}$$

$$z_1 = R_G + h_1 \tag{20}$$

$$\eta_1 = 0 \tag{21}$$

$$\theta_1 = \varepsilon_k \tag{22}$$

where $R_G$ is the Gaussian radius, an adequate approximation of the Earth radius

$$R_G = \frac{a^2 \cdot b}{(a \cdot \cos\varphi_1)^2 + (b \cdot \sin\varphi_1)^2} \tag{23}$$

with $a$ and $b$ as the semi-axes of the reference ellipsoid (e.g. GRS80). The z-axis connects the geocenter with the starting point, the y-axis is defined perpendicular to the z-axis in direction (azimuth angle) of the GNSS satellite in view. After setting the

initial parameters, the 'true' ray path is reconstructed iteratively by making use of ray-tracing shooting techniques. Therefore, total refractivity derived from an a priori field is read in and pre-processed for ray-tracing. Hereby, the input data is interpolated vertically and horizontally to the vertical plane, spanned by the y- and z-axis.

In the *ray-tracing loop*, for each height layer $h_{i+1}$ with $i = 1 : (t-1)$ whereby $t$ defines the top layer of the voxel model, the geocentric coordinates and the corresponding angles are computed as follows:

$$y_{i+1} = y_i + d_i \cdot \cos \varepsilon_i \tag{24}$$

$$z_{i+1} = z_i + d_i \cdot \sin \varepsilon_i \tag{25}$$

$$\eta_{i+1} = \arctan \frac{y_{i+1}}{z_{i+1}} \tag{26}$$

$$\theta_{i+1} = \arccos \left( \frac{n_i}{n_{i+1}} \cdot \cos(\theta_i + \eta_{i+1} - \eta_i) \right) \tag{27}$$

$$d_i = -(R_G + h_i) \cdot \sin \theta_i + \sqrt{(R_G + h_{i+1})^2 - (R_G + h_i)^2 \cdot \cos^2 \theta_i} \tag{28}$$

$$\varepsilon_{i+1} = \theta_{i+1} - \eta_{i+1} \tag{29}$$

where $d_i$ is the reconstructed path length between height layer $h_i$ and $h_{i+1}$ ($h_{i+1} > h_i$). It depends on the observation geometry but also on the atmospheric conditions (refractive indices $n_i$ and $n_{i+1}$). By default, for our analysis, the spacing between two height layers $h_i$ and $h_{i+1}$ was set to 5 m, which corresponds to a maximum path length $d_i$ of 100 m - assuming an elevation angle of 3° ($5m/sin3°$).

The ray-tracing loop stops when the ray reaches the top layer $t$ of the voxel model. Assuming spherical trigonometry, the spherical coordinates $(\varphi_{i+1}, \lambda_{i+1})$ of the ray path segments follow by Eqs. (24) and (25) to:

$$\varphi_{i+1} = \arcsin \left( \sin \varphi_1 \cdot \cos(\eta_{i+1} - \eta_1) + \cos \varphi_1 \cdot \sin(\eta_{i+1} - \eta_1) \cdot \cos \alpha \right) \tag{30}$$

$$\lambda_{i+1} = \lambda_1 + \arctan \left( \frac{\sin \alpha}{\cot(\eta_{i+1} - \eta_1) \cdot \cos \varphi_1 - \sin \varphi_1 \cdot \cos \alpha} \right) \tag{31}$$

where $\varphi_{i+1}$ and $\lambda_{i+1}$ are defined in the range $[-\pi/2, \pi/2]$ and $[-\pi, \pi]$, respectively. The ray coordinates are necessary for interpolation of the refractive indices $n_i$ and $n_{i+1}$ for the next processing step $i$ but also for computation of the intersection points with the voxel model boundaries.

The ray-tracing loop is repeated until $\varepsilon_t - \varepsilon_k + g_{bend}$ is smaller than a predefined threshold (e.g. $10^{-6}$ degrees). While the elevation angle $\varepsilon_t$ is obtained by Eq. (29) for $i = t - 1$, the correction term $g_{bend}$ accounts for the additional bending above the voxel model. Since atmosphere is almost in state of hydrostatic equilibrium, $g_{bend}$ can be well approximated by a bending model, like the one of Hobiger et al. (2008):

$$g_{bend}[°] = \frac{0.02 \cdot \exp^{\frac{-h}{6000}}}{\tan \varepsilon_k} \tag{32}$$

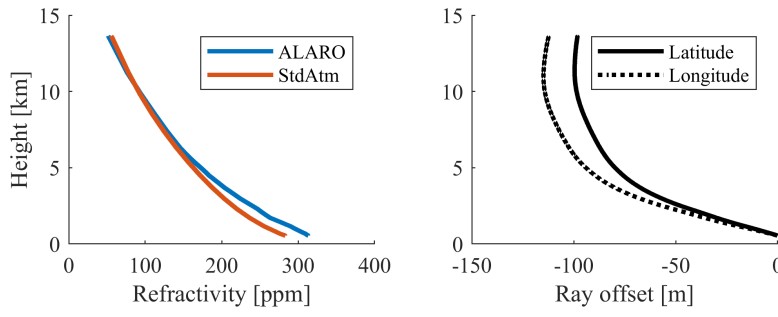

**Figure 2.** Ray-traced signal path differences (right) caused by differences in the a priori refractivity field (left)

where $h$ is replaced by $h_t$, the height of the voxel top layer. After convergence of the ray-tracing loop, the path length in each voxel is obtained by summing up the distances $d_i$ in each voxel. Thereby, allocation of the ray parts is carried out by comparison of the ray coordinates $(\varphi_i, \lambda_i, h_i)$ with the coordinates of the voxel model. The obtained ray paths in each voxel - for each station and each satellite in view - are used for setting up design matrix $\boldsymbol{A}$ (see Eq. 12).

## 4.2 Quality of reconstructed ray paths

### 4.2.1 The refractivity field

The quality of the ray-traced signal paths depends primarily on the quality of the refractivity field. Especially if no good a priori data can be made available, e.g. if standard atmosphere (StdAtm) is used instead of numerical weather model data (ALARO), the reconstructed signal path might deviate significantly from the 'true' signal path.

Figure 2 shows the impact of the refractivity field on the signal geometry, exemplary for a GNSS signal observed at station Jenbach, Austria ($\varphi = 47.4°, \lambda = 11.8°, h = 545m$) with $\varepsilon = 5°$ and $\alpha = 230°$. At this particular epoch (May 4, 2013, 15 UTC), standard atmosphere deviates by about 30 ppm from the ALARO model data. Assuming ALARO as reference, ray-tracing through standard atmosphere causes a ray deviation of 100-200 m (see Fig. 2, right).

In order to reduce the impact of possible refractivity errors on the reconstructed ray paths and in further consequence on the tomography solution, ray-tracing was carried out iteratively. Therefore, the refractivity field obtained from the first tomography solution replaces the initial refractivity field for ray-tracing for the next iteration and so on. The processing is repeated until $N_{\boldsymbol{w}}$ converges.

Figure 3 (left) shows the convergence behavior assuming standard atmosphere (StdAtm) and ALARO model data as input. In both cases, the standard deviation of the differences in path length between two consecutive epochs ($d_{k,i+1} - d_{k,i}$) was selected as convergence criteria. Both solutions converge after two iterations. Thereby, the path lengths within each voxel 'improve' by about $22m$ in case of standard atmosphere and by $11m$ in case of ALARO data. This result was expected, since ALARO data are closer to the 'true' atmospheric conditions. By comparison of Figure 3 (right) with Figure 2 (right) it is clearly visible,

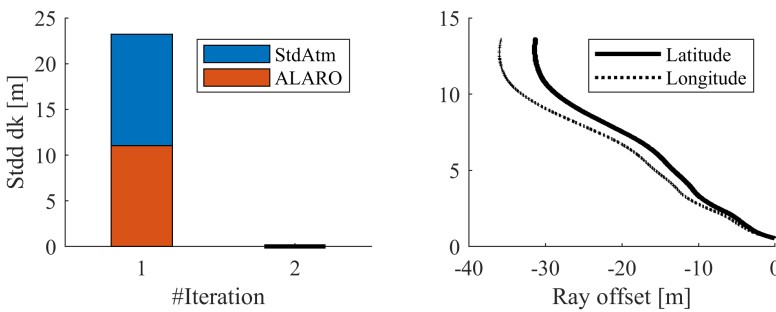

**Figure 3.** Convergence behavior (left) and ray-traced signal path differences after convergence (right). All iterations are based on the same first guess (standard atmosphere or ALARO numerical weather model data) but differ with respect to the refractivity field used for reconstruction of the bended signal paths

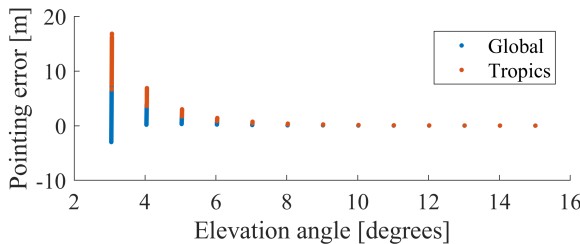

**Figure 4.** Point error at voxel model top $(h = 13.6km)$ caused by the bending model of Hobiger et al. (2008) - computed on a global $10°x10°$ grid over the period of one year, 2014 by comparison with ray-traced bending angles based on ECMWF analysis data

that the two additional iterations help to reduce the ray offset caused by errors in the standard atmosphere from $100 - 200m$ to $30 - 40m$. In Sect. 5 the resulting effect on the tomography solution is assessed.

### 4.2.2 The empirical ray-bending model

Besides the refractivity field, the quality of the reconstructed ray paths might be also affected by errors in the bending model as defined by Eq. (32). Comparisons of the bending model with ray-traced bending angles on a global $10°x10°$ grid over the period of one year reveal that the error in bending is usually kept below 0.8 arcsec. Assuming a GNSS site near sea-level and an elevation angle of $5°$, an error in bending angle of $±0.8$ arcsec causes an error in path length of up to $±10m$, i.e. the reconstructed GNSS signal enters the voxel model slightly earlier or later than the observed GNSS signal. In Fig. 4, the bending error is visualized as pointing error at voxel model top. However, for the tomography solution this effect is too small to be significant. Thus, it can be concluded that the bending model of Hobiger et al. (2008) is well applicable for reconstruction of the bending angle above the voxel model, in particular if the voxel model height $h_t$ is set to 12 km or higher.

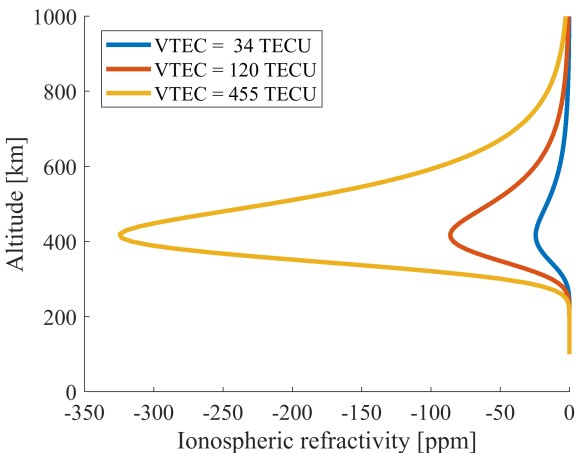

**Figure 5.** Profiles of ionospheric refractivity $N(f)$ assuming signal-frequency $f = 1575.42MHz$

### 4.2.3 Ionospheric bending effects

Beyond, also the ionosphere influences GNSS signal propagation. In order to assess the impact of free electrons in the iono-sphere (above 80 km altitude) on the signal path, the electron density model by Anderson et al. (1987) was executed in three scenarios, assuming a vertical total electron content $(VTEC = \int N_e \cdot dh)$ of 34 TECU (average daytime), 120 TECU (solar
maximum) and 455 TECU (maximum possible, see Wijaya, 2010), respectively.

$$N(f) = 10^6 \cdot \frac{-40.2993 \cdot N_e}{f^2} \tag{33}$$

By making use of Eq. (33), the obtained electron density profiles were converted into profiles of refractivity (N), assuming signal frequency $f_1 = 1575.42MHz$ (GPS L1) and $f_2 = 1227.60MHz$ (GPS L2). Figure 5 shows the obtained vertical profiles of ionospheric refractivity, exemplary for frequency $f_1$. The higher the signal-frequency $f$ the lower the phase velocity through
the ionosphere and the less is its refraction. Following the approach by Wijaya (2010), the ray paths in the ionosphere were reconstructed, separately for GPS L1 and L2. The analysis revealed significant path differences between the 'true' ray path and its chord line but also between the two signal-frequencies. Assuming a VTEC of 455 TECU and an elevation angle of $3°$, the maximum deviation from the straight line signal path is 800 m for L1 and 550 m for L2 respectively, at $h = 400km$, slightly below the layer of peak electron density. Fortunately, ray path deviation decreases significantly with decreasing VTEC
and altitude to a few tens of meters at $h = 13.6km$ (the upper rim of the troposphere at which the top of the voxel model was defined). In consequence, the impact of free electrons on the signal path in the lower atmosphere is negligible under moderate and low ionospheric conditions.

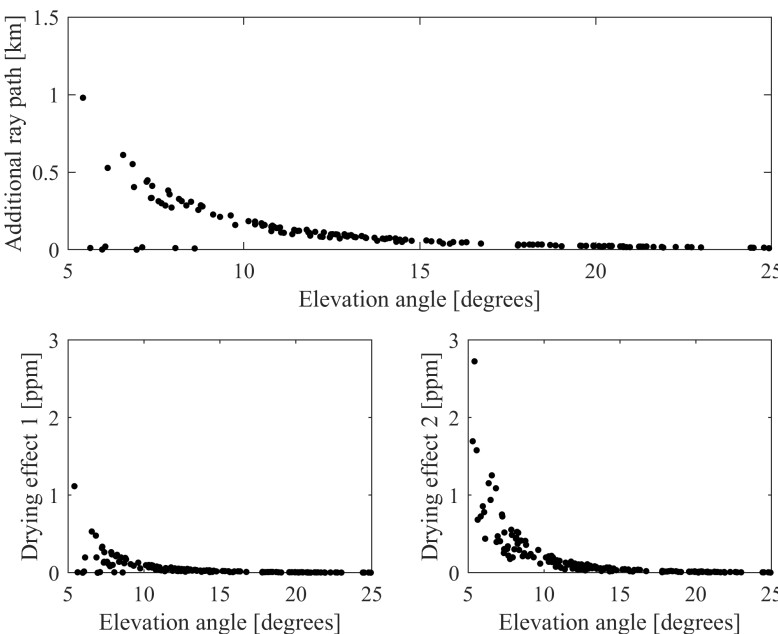

**Figure 6.** Additional ray path caused by straight line assumption (top), the resulting drying effect due to the additional ray path (bottom left) and the resulting drying effect caused by the fact that the straight line ray travels through lower atmospheric levels than the 'true' bended ray (bottom right)

## 5 Impact of atmospheric bending on the tomography solution

In the following, the differences between straight line and bended ray-tracing are further analyzed. For highest consistency, the ray-tracing approach defined in Sect. 4 was used for both, straight line and bended ray-tracing. The only difference is that in case of straight line ray-tracing the ratio $n_i/n_{i+1}$ in Eq. (27) was set to '1'. Thereby, it can be guaranteed that only the impact of atmospheric refraction is assessed.

### 5.1 Expected drying effect

In the beginning, the ray position is equal for both methods but diverges with increasing height. Thereby, the bended ray is travelling in most cases 'above' the straight ray, i.e. the straight ray enters the voxel model top 'earlier' than the bended ray. This leads to the effect that the straight ray remains longer in the voxel model than the bended ray, i.e. the straight ray path within the voxel model ($h < 13.6km$) is systematically longer than the bended ray path. The differences between both ray paths are plotted in Fig. 6 (top) as function of elevation angle. Therefore, ALARO model data was selected as input for the bended ray-tracer.

The additional ray path decreases rapidly with increasing elevation angle. Thus, a *mixed ray-tracing* approach can be defined, which considers ray bending only for $\varepsilon \leq 15°$. Beyond, the additional ray path is below 0.1 km, and straight line ray-tracing is sufficient for ray path reconstruction.

Figure 6 (top) shows also that in some cases even for low elevation angles the difference in path length is small (below 0.1 km). This appears when the ray enters the voxel model not through the top layer but through a later surface of the voxel model. In this particular case, the difference in path length between both ray-tracing approaches is negligible (be aware that only the entire distance through all voxels is comparable for both ray-tracing approaches but not the individual distances in each voxel). Figure 6 (bottom) shows the expected drying effects in the tomography solution caused by errors in the reconstructed signal paths assuming straight line geometry. Hereby, it is distinguished between the drying effect caused by the additional ray path ($dNw_1$) and the drying effect caused by the fact that the straight line travels through lower layers of the voxel model ($dNw_2$). Both effects were assessed as follows:

$$dNw_1 = SWD_b \cdot (d_{k,s} - d_{k,b}) \tag{34}$$

$$dNw_2 = (SWD_b - SWD_s) \cdot d_{k,s} \tag{35}$$

whereby $SWD_b$ and $SWD_s$ are the slant wet delays obtained by ray-tracing through ALARO model data along the bended and the straight line ray path, respectively. The variables $d_{k,b}$ and $d_{k,s}$ are the corresponding path lengths within the voxel model. The sum of their differences along the ray paths are identical with the additional ray paths plotted in Figure 6 (top).

Both drying effects have to be considered as additive and are strongly connected to the current atmospheric conditions as well as to the parametrization applied for interpolation of the refractivity field. In our analysis we assumed an exponential decrease of refractivity between the vertical layers of the voxel model and applied a bi-linear interpolation method for horizontal interpolation between the grid points.

## 5.2   Results from the Austrian GNSS tomography test case

In order to study the impact of bended ray-tracing on the tomography solution, a GNSS tomography test case was defined. The corresponding settings are summarized in Table 1.

Figure 7 (left) shows the differences in wet refractivity between Sol1 and Sol2 (as defined in Table 1). Even though on average over all voxels no bias in wet refractivity is observed, specific voxels show differences in wet refractivity of up to $10ppm$, particularly if due to bending different voxels than in the straight line solution are traversed.

Figure 7 (right) shows the differences in wet refractivity between the first two iterations of the mixed ray-tracing approach (Sol2). In this particular case, refractivity differences are smaller than $0.05ppm$, which implies that the a priori model used for ray-tracing is already close to the 'true' atmospheric conditions, i.e. in this particular case no further iteration was necessary. From all differences in wet refractivity over 248 epochs in May 2013, a maximum of $14.2ppm$, a bias of $0.12ppm$ and a standard deviation of $0.24ppm$ were obtained. Although the bias and standard deviation over all voxels is small, differences of

**Table 1.** Summary of GNSS tomography test case settings

| Parameter | Settings |
|---|---|
| Period | May 2013, 8 epochs per day |
| Voxel domain | Western Austria $(46.4 - 48.0°$ lat, $10.4 - 13.4°$ lon, $h = 0 - 13.6 km)$ |
| Voxel size | $0.4°$ lat x $0.6°$ lon (4 x 5 ground voxels), 15 height layers |
| GNSS data | 30 sec dual-frequency GPS and GLONASS observations - obtained from 6 |
| | EPOSA reference sites: SEEF, MATR, JENB, KIBG, ROET, SILL |
| A priori model | ALARO analysis data of temperature and specific humidity |
| | - provided on 18 pressure levels in grib1 format for 8 epochs per day |
| Observations | SWDs for all GPS and GLONASS satellites in view above $3°$ |
| | elevation angle - derived from 1h ZTD and 2h gradient estimates |
| Ray-tracer | Sol1: Straight line ray-tracing for all observations up to $h = 13.6 km$ |
| | Sol2: Straight line ray-tracing for $\varepsilon > 15°$ and bended |
| | ray-tracing for $\varepsilon \leq 15°$ (mixed approach) up to $h = 13.6 km$ |

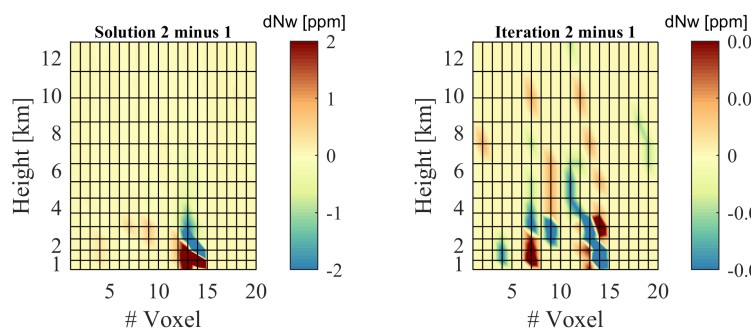

**Figure 7.** Error in wet refractivity caused by straight line assumption (left) and the differences in wet refractivity between first and second iteration (right). Thereby, voxel number '1' is dedicated to the South-West corner and number '20' to the North-East corner of the voxel model. For visualization, a bi-linear interpolation method was applied between the grid points. Analyzed period: May 4, 2013, 15 UTC

about $1 ppm$ were observed on average at each epoch, especially when observations below 10 degrees elevation angle enter the tomography solution.

## 5.3   Validation with radiosonde data

For validation of the mixed ray-tracing approach against straight line ray-tracing, the tomography derived wet refractivity fields were compared with radiosonde data at the airport of Innsbruck $(\varphi_i = 47.3°, \lambda = 11.4°, h = 579 m)$. First, the radiosonde data obtained once a day between 2 and 3 UTC were pre-processed, i.e. outliers in temperature were removed and dew point temperature was converted to water vapor pressure and further to wet refractivity. Finally, the radiosonde profiles were vertically

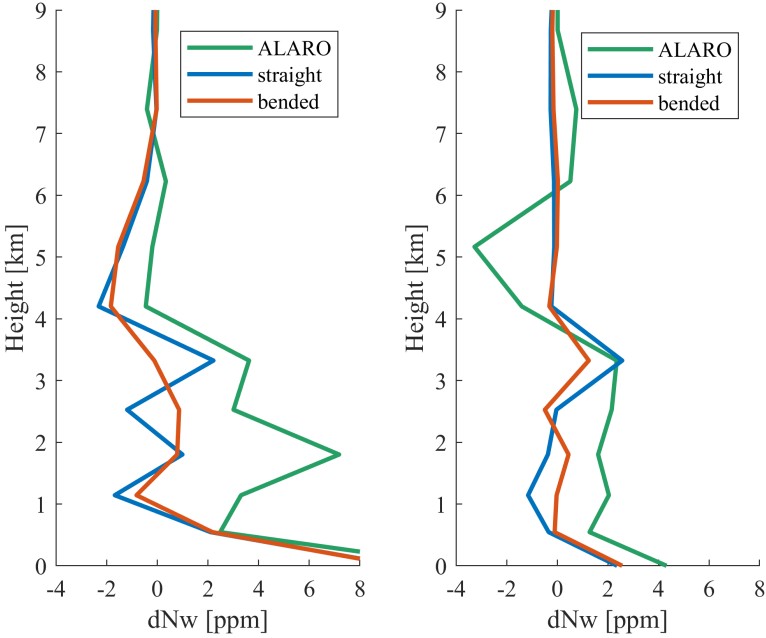

**Figure 8.** Differences in wet refractivity between radiosonde, ALARO and the two tomography solutions based on straight line (blue) and bended ray-tracing (red), exemplary for May 1, 2013, 3 UTC (left) and May 31, 3 UTC (right)

interpolated to the height layers of the voxel model and the tomography derived wet refractivity fields were horizontally interpolated to the ground-position of the radiosonde launching site, respectively. Figure 8 shows the differences in wet refractivity as function of height above surface, exemplarily for two epochs in May, 2013. In both cases, the bended ray-tracing approach helps to reduce the tomography error by about $1-2ppm$, especially in the lower 4 km of the atmosphere. Largest differences are

5   visible when the bended ray traverses other voxels than its chord line. This appeared in about 2 % of the test cases, especially if observations below 10 degrees elevation angle enter the tomography solution.

## 6   Conclusions

GNSS signals which enter the neutral atmosphere at low elevation angles ($\varepsilon < 15$ degrees) are significantly affected by atmospheric bending. In case the bending is neglected when setting up design matrix $\boldsymbol{A}$, a systematic error of up to $10-20ppm$,

10   on average of $1-2ppm$ is introduced into the GNSS tomography solution. This error can be widely reduced if atmospheric bending is considered in reconstruction of the signal paths. Therefore, a 2D piecewise linear ray-tracing approach was defined, which describes the bended GNSS signal path by small line segments. By limiting the length of the line segments to $100m$ in case of $\varepsilon = 3°$ or even shorter for higher elevation angles, the 'true' signal path can be widely reconstructed. However, the quality of the reconstructed signal paths depends primarily on the quality of the a priori refractivity field. Comparisons

between refractivity fields derived from standard atmosphere and ALARO weather model data reveal that a refractivity error of $30ppm$ can cause a ray deviation of up to several hundred meters, i.e. the distance traveled in each voxel but also the number of traversed voxels is prone to misallocations. In consequence, reliable a priori data, e.g. derived from numerical weather model data, are recommended for GNSS tomography.

Nevertheless, if reliable a priori data are not available or if the quality is unknown, iterative ray-tracing helps for reducing the impact of wet refractivity errors on the tomography solution. Therefore, the wet refractivity field obtained from an initial tomography solution is used for reconstruction of the signal paths for the next iteration. The processing is repeated until the tomography solution converges. This ensues usually after two iterations. Beyond, a bending model, like the one provided by Hobiger et al. (2008) helps to significantly reduce computational cost by describing the remaining bending in the higher

atmosphere (above the voxel model). In consequence, the ray-tracer can be stopped right after the reconstructed signal leaves the voxel model. In case of $h_t = 13.6km$, the number of processing steps is reduced by $85\%$, which is a tremendous reduction in processing time without significant loss of accuracy.

     In contrast, ionospheric bending effects have less impact on the GNSS tomography solution. Even during periods of solar maximum, ray path deviation caused by ionospheric bending is negligible for signals in L-band $(1-2GHz)$. However, even if

ionospheric bending has no impact on the tomography solution, first and higher order ionospheric effects should be taken into account when processing GNSS phase observations.

     Besides, comparisons with radiosonde data revealed that if atmospheric bending effects are considered in GNSS tomography, the quality of the tomography solution can be improved by $1-2ppm$. Within the defined test case, especially voxels in the lower $4km$ of the atmosphere benefitted from the applied mixed ray-tracing approach. Due to significant optimization, the

mixed ray-tracing approach ensures processing of large tomography test cases in adequate time. A test case with 72 GNSS sites and 7 x 9 x 15 voxels can be processed in less than two minutes. Thus, the developed mixed ray-tracing approach is applicable also in near real-time and therefore well suited for operational purposes.

*Code availability.* The 2D piecewise linear ray-tracer for GNSS tomography as well as the RADIATE ray-tracer are part of the Vienna VLBI and Satellite Software (VieVS). The code of the RADIATE ray-tracer is available at https://github.com/TUW-VieVS/RADIATE. For more

details to VieVS, the reader is referred to http://vievswiki.geo.tuwien.ac.at.

## Appendix A: Unmodelled bending effects in the Vienna hydrostatic mapping function

In case of VMF1 (Böhm et al., 2006) or similar mapping concepts, azimuthal asymmetry is not considered and for convenience, only a single hydrostatic mapping coefficient per site $(a_h)$ is determined as follows:

$$a_h = -\frac{mf_h(\varepsilon) \cdot \sin\varepsilon - 1}{\frac{mf_h(\varepsilon)}{\sin\varepsilon + \frac{b_h}{\sin\varepsilon + c_h}} - \frac{1}{1 + \frac{b_h}{1 + c_h}}}. \tag{A1}$$

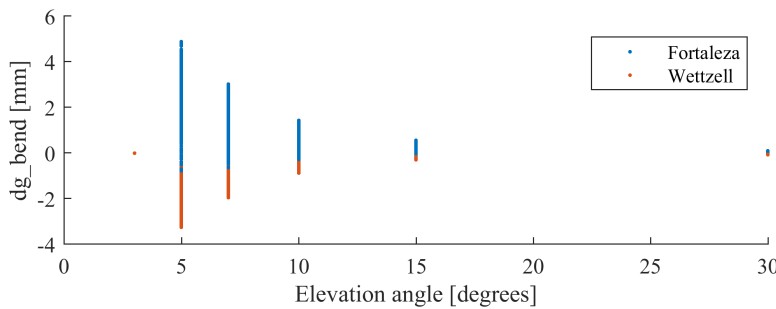

**Figure A1.** The unmodeled geometric bending effect in VMF1 hydrostatic mapping function ($dg_{bend}$), exemplary for VLBI sites Fortaleza, Brazil and Wettzell, Germany. Analyzed period: Jan-Feb 2014

where $b_h$ is 0.0029, $c_h$ depends on the day of year and latitude and $mf_h(\varepsilon)$ is defined as the ratio between $SHD(3°)$ and $ZHD$, obtained by ray-tracing through numerical weather model data. For assessing the remaining unmodeled geometric bending $dg_{bend}(\varepsilon, \alpha)$, ray-traced slant hydrostatic delays were compared with 'mapped' slant hydrostatic delays as follows:

$$dg_{bend}(\varepsilon, \alpha)[m] = ZHD[m] \cdot mf_h(\varepsilon) - ZHD[m] \cdot mf_{h0}(\varepsilon) - g_{bend}(\varepsilon, \alpha)[m] \tag{A2}$$

where $ZHD$ is the zenith hydrostatic delay obtained by vertical integration, $g_{bend}(\varepsilon, \alpha)$ is the geometric bending effect as obtained by ray-tracing, $mf_h(\varepsilon)$ is the VMF1 hydrostatic mapping function determined by $SHD(3°)/ZHD$ and $mf_{h0}(\varepsilon)$ is the hydrostatic mapping function determined by $SHD_0(3°)/ZHD$, whereby $SHD(3°)$ and $SHD_0(3°)$ are the slant hydrostatic delays obtained by ray-tracing for an vacuum elevation angle $\varepsilon_k = 3°$ with and without geometric bending, respectively. Figure A1 shows the remaining unmodeled geometric bending as obtained for six elevation angles (and 16 equidistant az-

imuth angles), exemplary for the two VLBI sites Fortaleza, Brazil ($\varphi = -3.9°, \lambda = 321.6°, h = 23m$) and Wettzell, Germany ($\varphi = 49.1°, \lambda = 12.9°, h = 669m$). In case of $\varepsilon = 3°$, almost no bending error is visible since $mf_h(\varepsilon)$ was tuned for this elevation angle. However, for other elevation angles, the unmodeled geometric bending is about 3 % of the slant hydrostatic delay, e.g. up to $\pm 5mm$ at $5°$ elevation angle. In case of Wettzell, $dg_{bend}(\varepsilon, \alpha)$ is mostly negative, i.e. the 'mapped' $SHD$ is smaller than the observed $SHD$ and vice versa for Fortaleza. So far, these small variations are neglected when using VMF1

hydrostatic mapping function in GNSS signal processing.

*Acknowledgements.* Open access funding was provided by Austrian Science Fund (FWF). The authors would like to thank the Austrian Science Fund (FWF) for financial support of this study within the project RADIATE ORD (ORD 86) and the Austrian Research Promotion Agency (FFG) for finacial support within the project GNSS-ATom (840098).

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
