# Peer review of "Atmospheric bending effects in GNSS tomography"

_Atmospheric Measurement Techniques, 2018_

## Referee Comment (RC1) · Anonymous Referee #1 · 14 Aug 2018

General comments:

The paper studies bending effects in GNSS tomography. I recommend to consider the paper for publication. However, the following points must be addressed.

Specific comments:

(1) I recommend to rewrite the 'Introduction'. The section 'Introduction' must be more general. In the 'Introduction' there is no need for technical details and formulas. Technical details and formulas must be provided in the following section (see next point). Instead, provide a brief overview on the state of the art in tomography. Provide some relevant references, e.g. Bender et al., 2011, Champollion et al., 2005, Hirahara, 2000, and their findings. In all the above mentioned works bending effects were ig-

nored. Therefore you should then provide references where bending effects are taken into account. Here you should mention Zus et al (2015) (not cited in the manuscript) and a paper that appeared two years later Aghajany and Amerian (2017) (cited in the manuscript).

(2) Section 1 ('Introduction') and 2 ('Atmospheric bending effects in GNSS signal processing') need a complete makeover. I suggest to merge the two sections to one section with the following title 'Atmospheric bending effects and WV tomography'. To my understanding you are concerned with SWDs and not STDs. In short, i recommend the following structure for this section:

2.1 Atmospheric bending effects

Here you should at first introduce the basic observable, i.e., STDs. You can either use eq 2 or 10. They are essentially the same. I recommend to use eq 2. Hence, you start as follows: The STD is defined as (Bevis et al. 1992)

STD = int n ds – g

n...index of refraction s...arc length of bent ray-path (refer to section 3) g...geometric distance between satellite and station

Then, you introduce refractivity N. In essence

n=10**(-6) N + 1

and therefore

STD = int 10**(-6) N ds + s – g

Then you introduce the hydrostatic and wet refractivity

N= Nh + Nw

and therefore

STD = int 10**(-6) Nh ds + s – g + int 10**(-6) Nw ds

Next, you introduce the following quantities

$SHD = \int 10^{**}(-6) \text{ Nh ds} + s - g$

and

$SWD = \int 10^{**}(-6) \text{ Nw ds}$

such that

$STD = SHD + SWD$

At this point it is again important to mention that the ray-path (and therefore the arc-length s) depends on the 'total' refractivity N (refer to section 3).

Then you claim that the SWD can be accurately estimated with the GNSS. In essence, you introduce the assembled STD that is used in the GNSS analysis (eq 11)

$STD\_GPS = ZHD\_GPS * mh(e) + ZWD\_GPS * mw(e) + mg(e) (N \cos(a) + E \sin(a))$

and provide the formula that you use to recover the SWD. I can only guess (please provide the details) something like this

$SWD\_GPS = STD\_GPS - ZHD\_NWM * mh(e)$

or better yet something like this

$SWD\_GPS = STD\_GPS - ZHD\_NWM * mh(e) - mg(e) (Nh\_NWM \cos(a) + Eh\_NWM \sin(a))$

where ZHD_NWM is ZHD derived from a NWM (or derived from in situ pressure sensor) and Nh_NWM and Eh_NWM are the hydrostatic gradient components derived from a NWM. Here you can mention that the hydrostatic mf (which is derived under the assumption of a spherically layered troposphere) takes into account the geometric bending term. In essence,

$mh = (\int 10^{**}(-6) \text{ Nh ds} + s - g) / ZHD$

With this details you are finished with 2.1 and prepared for 2.2

2.2 WV tomography

Since the observable you consider are SWDs, there is no need for eq 3 and 4. You can start directly with the following formula

SWD = int 10**(-6) Nw ds

and its numerical approximation

SWD $\sim$ sum_i 10**(-6) Nw_i ds_i

where you again explicitly mention that, because of ray-path bending, s does not equal g (refer to section 3). Then you can proceed with your eq 6 and 7. It is important that you explain what P and Pc is. I guess (please provide the details) that Pc tells us something about the uncertainty of the observations and P tells us something about the uncertainty of the a-prior (first-guess or background) wet refractivity?

With this you are finished with section 2 and proceed with your section 3.

(3) I suggest that somewhere in the manuscript you plot the following difference

dSWD = SWD_T - SWD_0

as a function of the elevation angle for some station. Here SWD_0 is the SWD calculated along the straight line path and SWD_T is the SWD calculated along the ray-path. In essence,

dSWD = sum_i 10**(-6) Nw_i ds_i - sum_i 10**(-6) Nw_i dg_i

I guess you will find that the following inequality holds true for any elevation angle

SWD_0 > SWD_T

due to the fact that the ray-path traverses the troposphere at higher altitudes than the straight line path. This would imply that when ray-path bending is not taken into account

in tomography the reconstructed troposphere is too dry. To see this you could chose some 'true' wet refractivity field, say N_w0, and simply replace in your eq 7 the term

(SWD - A*N_w0)

by

dSWD. I strongly recommend to do this somewhere in the manuscript.

technical corrections:

Abstract:

L7: '...Thereby, the ray-tracing approach itself but primarily the quality of the a prior field has a significant impact on the reconstruction quality...' improve the writing.

Introduction:

L15: 'GNSS' abbreviation not introduced here.

Section 3.1

L24: What is the 'outgoing' elevation angle? Please provide a clear definition here.

L18: The inner loop you use is to solve the so called 'homing in problem' (you make use of a shooting method). Please state this more clearly here.

Section 3.2

L4: '...In consequence, the reconstructed...'. The phrase 'In consequence' can be avoided here and at various other places.

Section 3.2.2

I suggest to show in Fig 4. directly the difference in SWD[m] and not the difference in the bending angle[arcsec]. Also, i do not find the formula for the bending angle in the manuscript. I guess you mean something like arcos(v1,v2) where v1 is the tangent unit vector of the ray-path at the satellite and v2 is the tangent unit vector of the ray-path at

the station?

Section 4.1

I suggest to add in Fig 8. the difference for the a-prior (first guess or background) refractivity (ALARO). Is the radiosonde data assimilated into ALARO?

References:

Check all references carefully. For example,

Fritsche, M., Dietrich, R., Knofel, C., Rulke, A., and Vey, S.: Impact of higher-order ionospheric terms on GPS estimates, Geophys. Res. Lett., 32, 1–5, 2005.

Bender, M., Stosius, R., Zus, F., Dick, G., Wickert, J., Raabe, A. (2011): GNSS water vapour tomography – Expected improvements by combining GPS, GLONASS and Galileo observations. - Advances in Space Research, 47, 5, pp. 886—897. DOI: http://doi.org/10.1016/j.asr.2010.09.011

In the manuscript the correct citation should be e.g. Böhm et al 2006 and not Böhm et al 2006a. Likewise the correct citation should be Hobiger et al 2008 and not Hobiger et al 2008a (there is no b).

Additional Reference:

Zus, F., Dick, G., Heise, S. and Wickert, J.: A forward operator and its adjoint for GPS slant total delays, Radio Science, 50, 393– 405, doi: 10.1002/2014RS005584, 2015.

---

## Referee Comment (RC2) · Anonymous Referee #2 · 5 Oct 2018

The paper outlines a method for including ray bending in GNSS tomography. My main concern relates to the iterative retrieval technique, and the possible confusion between "a priori" and "first guess". Equation 7 should not be used in iterative form, and therefore a solution from eq. 7 should not be used as "a priori" for the next iteration (lines 7-8, page 7). This is covered in section 5.6.2 ("A popular mistake") in Inverse Methods for Atmospheric Sounding Theory and Practice by Clive Rogers. I suggest that this issue should be clarified before moving to the discussion phase, because it impacts all of the results.

Minor points.

I do not understand why the authors use singular value decomposition in Eq. 7. Normal matrix inversion should suffice. The meaning of the weighting matrices should ex-

plained (they are the inverse of covariance matrices I think) and they should be clearly defined.

The work might benefit from investigating how ray-bending is handled in GPS radio occultation measurements. EG, Burrows, C. P., Healy, S. B., and Culverwell, I. D.: Improving the bias characteristics of the ROPP refractivity and bending angle operators, Atmos. Meas. Tech., 7, 3445-3458, https://doi.org/10.5194/amt-7-3445-2014, 2014.

---

## Author Comment (AC1) · 16 Nov 2018

We would like to thank referee #2 for his/her valuable suggestions. In the following, you will find our responses, separately for each comment/concern. We are confident, that we can provide a revised version of the manuscript, which addresses all of your points of the first review and looking forward to your positive feedback.

Comment: The paper outlines a method for including ray bending in GNSS tomography. My main concern relates to the iterative retrieval technique, and the possible confusion between "a priori" and "first guess".

Author's response: In the revised version of the manuscript, the terms "a priori" and "first guess" will be used according to its definition in 'Inverse Methods for Atmospheric

[Figure]

Sounding Theory and Practice' by Clive Rogers, section 5.6.2 as "the best estimate of the state before the measurement is made" (a priori) and as 'the starting point of an iteration' (first guess).

Comment: Equation 7 should not be used in iterative form, and therefore a solution from eq. 7 should not be used as "a priori" for the next iteration (lines 7-8, page 7). This is covered in section 5.6.2 ("A popular mistake") in Inverse Methods for Atmospheric Sounding Theory and Practice by Clive Rogers. I suggest that this issue should be clarified before moving to the discussion phase, because it impacts all of the results.

Author's response: We agree, by using the result of a previous iteration as first guess for the next iteration provides in a least squares sense not an optimal estimation. In consequence, we adapted our approach slightly – following the non-linear iterative approach suggested by Iyer and Hirahara, Seismic tomography: Theory and practice, 1993. Therefore, we keep the initial a priori field as first guess for each iteration but use the result of each iteration for improving the reconstruction of the signal paths. Therewith, and according to Fermat's principle (first order changes of the ray path lead to second order changes in travel time) the estimation error can be significantly reduced. Besides, the aim should be always to make use of the best available a priori field, especially if low elevation observations are involved. If this is not possible, the proposed non-linear iterative approach will allow for a more exact reconstruction of the signal paths, even if the used a priori refractivity field deviates significantly from the true atmospheric conditions.

Author's changes in manuscript: Due to the adaption in the tomography approach, several parts of the manuscript will be revised. This includes: Section 3.2.1 The a priori refractivity field, page 7 line 7-14, Section 4 Impact of atmospheric bending on the tomography solution, page 10 line 12-22, Section 4.1 Validation with radiosonde data, page 12 line 1-3, Section 5 Conclusions, page 12 line 6-7, page 13 line 4-7 and line 16-18 and Figure 3, 7 and 8.

Minor points Comment: I do not understand why the authors use singular value decomposition in Eq. 7. Normal matrix inversion should suffice.

Author's response: Solving Eq. (6) for Nw requires the inversion of matrix A. In GNSS tomography, matrix A is mostly singular; in consequence, a straightforward inversion is not possible. Thus, for our solution we make use of singular value decomposition methods for solving the ill-posed inversion problem. Together with a proper singular value selection method like L-curve technique (see Moeller, 2017), it allows for solving the equation system and for retrieving as much signal as possible from the observations without introducing too large artefacts.

Author's changes in manuscript: We will add a more detailed explanation of our approach, in particular why we used SVD, to the revised version of the manuscript below Eq. 7.

Comment: The meaning of the weighting matrices should explained (they are the inverse of covariance matrices I think) and they should be clearly defined.

Author's response: The weighting matrices P and Pc are the inverse of the variance-covariance matrices C and Cc. They are defined separately for the SWDs, i.e. the observations (C) and the first guess (Cc). The diagonal elements of C were computed as function of elevation angle: $sin(e)^2 \ast sig\_ZTD^2$, whereby sig_ZTD = 2.5 mm reflects the accuracy of the estimated zenith total delays. The diagonal elements of Cc were derived from a weighting model based on height-dependent error curves for pressure, temperature and specific humidity in form of standard deviations (see Steiner A. K. et al., Error characteristics of refractivity profiles retrieved from CHAMP radio occultation data, 2006). Both error matrices were introduced into the tomography solution for proper weighting of the individual observations against the first guess.

Author's changes in manuscript: An explanation of the weighting method will be added to Chapter 1 after Eq. 7.

Comment: The work might benefit from investigating how ray-bending is handled in GPS radio occultation measurements. EG, Burrows, C. P., Healy, S. B., and Culverwell, I. D.: Improving the bias characteristics of the ROPP refractivity and bending angle operators, Atmos. Meas. Tech., 7, 3445-3458, https://doi.org/10.5194/amt-7-3445-2014, 2014.

Author's response: According to the reviewer's suggestion, we did a literature study on ray-tracing methods, in particular for assimilation of radio occultation measurements into numerical weather prediction systems.

While wave-theoretic approaches can help to increase the vertical resolution of the refractivity profile derived from radio occultation measurements, nowadays approaches based on the principles of geometric optics are still valid for operational analysis of radio occultation measurements. The main reason is that necessary assumption in signal processing, like the symmetric assumption or limitations in the GPS signal structure are still the dominating factors (Melbourne W. G., Radio Occultation Using Earth Satellites: A Wave Theory Treatment, 2004). Thus, and for highest consistency, also ray-tracing approaches based on the principles of geometric optics are widely used for reconstruction of the signal geometry, especially for assimilation of radio occultation measurements.

In contrast to radio occultation, the analysis of ground-based observations, including the weighting of ground-based observations in GNSS data processing, is based on different geometrical parameters. While in radio occultation, the bending angle is described as function of impact parameter a, for ground-based observations, elevation and azimuth angle are used for characterizing the observation geometry. In consequence, the optimal ray-tracing approach for ground-based observation and its conversion differs from approaches used for radio occultation measurements.

Nevertheless, in both cases the quality of the a priori field, but also how refractivity is interpolated between the given model levels, will have an impact on the ray-traced

signal paths. Latter is clearly shown in the paper of Burrows C. P. et al. (2014), as suggested by the reviewer.

In the best case, the interpolation between model levels is carried out on state parameter level, i.e. separately for pressure, temperature and water vapour pressure as obtained from the a priori field. However, since this approach is not applicable in iterative tomography processing, we assume exponentially varying refractivity. We are aware, that thereby the bending angle is slightly underdetermined. Nevertheless, in relation to the total bending effect, its contribution is fractional. Thus, we will not further modify our ray-tracing approach but we will address this affect in section '5 Conclusions'.

Author's changes in manuscript: In the manuscript, we will add a brief review of existing ray-tracing approaches in section '3 Reconstruction of GNSS signal paths' and will highlight the differences between radio occultation and ground-based observations and its consequences on the ray-tracing approach. Furthermore, we will add a paragraph to section '5 Conclusion' to address the interpolation problem in the iterative non-linear tomography approach to highlight its impact on the tomography solution.

---

## Author Comment (AC2) · 16 Nov 2018

We would like to thank referee #1 for his/her detailed comments on our manuscript. In the following, you will find our responses to each specific comment and technical correction addressed. Looking forward to your positive feedback. A revised version of the manuscript will follow.

Specific comment (1): I recommend to rewrite the 'Introduction'. The section 'Introduction' must be more general. In the 'Introduction' there is no need for technical details and formulas. Technical details and formulas must be provided in the following section (see next point). Instead, provide a brief overview on the state of the art in tomography. Provide some relevant references, e.g. Bender et al., 2011, Champollion et al., 2005,

[Figure]

Hirahara, 2000, and their findings. In all the above mentioned works bending effects were ignored. Therefore you should then provide references where bending effects are taken into account. Here you should mention Zus et al (2015) (not cited in the manuscript) and a paper that appeared two years later Aghajany and Amerian (2017) (cited in the manuscript).

Author's response: According to the reviewer's suggestion, we will restructure the first section of the manuscript as follows: The technical details and formulas will be shifted to a new section '3 The principles of GNSS tomography'. A paragraph about achievements in GNSS tomography will be added and all references suggested by the reviewer will be mentioned in the revised version of the manuscript.

Specific comment (2): Section 1 ('Introduction') and 2 ('Atmospheric bending effects in GNSS signal processing') need a complete makeover. I suggest to merge the two sections to one section with the following title 'Atmospheric bending effects and WV tomography'. To my understanding you are concerned with SWDs and not STDs. In short, I recommend the following structure for this section: 2.1 Atmospheric bending effects Here you should at first introduce the basic observable, i.e., STDs. You can either use eq 2 or 10. They are essentially the same. I recommend to use eq 2. Hence, you start as follows: The STD is defined as (Bevis et al. 1992) STD = int n ds − g n...index of refraction s...arc length of bent ray-path (refer to section 3) g...geometric distance between satellite and station Then, you introduce refractivity N. In essence n=10*(-6) N + 1 and therefore STD = int $10^{**}$(-6) N ds + s − g Then you introduce the hydrostatic and wet refractivity N= Nh + Nw and therefore STD = int $10^{**}$(-6) Nh ds + s − g + int $10^{**}$(-6) Nw ds Next, you introduce the following quantities SHD = int $10^{**}$(-6) Nh ds + s − g and SWD = int $10^{**}$(-6) Nw ds such that STD = SHD + SWD At this point it is again important to mention that the ray-path (and therefore the arclength s) depends on the 'total' refractivity N (refer to section 3). Then you claim that the SWD can be accurately estimated with the GNSS. In essence, you introduce the assembled STD that is used in the GNSS analysis (eq 11) STD_GPS = ZHD_GPS

* mh(e) + ZWD_GPS * mw(e) + mg(e) (N cos(a)+ E sin(a)) and provide the formula that you use to recover the SWD. I can only guess (please provide the details) something like this SWD_GPS = STD_GPS - ZHD_NWM * mh(e) or better yet something like this SWD_GPS = STD_GPS - ZHD_NWM * mh(e) - mg(e) (Nh_NWM cos(a)+ Eh_NWM*sin(a)) where ZHD_NWM is ZHD derived from a NWM (or derived from in situ pressure sensor) and Nh_NWM and Eh_NWM are the hydrostatic gradient components derived from a NWM. Here you can mention that the hydrostatic mf (which is derived under the assumption of a spherically layered troposphere) takes into account the geometric bending term. In essence, mh = (int 10**(-6) Nh ds + s − g) / ZHD With this details you are finished with 2.1 and prepared for 2.2

2.2 WV tomography Since the observable you consider are SWDs, there is no need for eq 3 and 4. You can start directly with the following formula SWD = int 10**(-6) Nw ds and its numerical approximation SWD _ sum_i 10**(-6) Nw_i ds_i where you again explicitly mention that, because of ray-path bending, s does not equal g (refer to section 3). Then you can proceed with your eq 6 and 7. It is important that you explain what P and Pc is. I guess (please provide the details) that Pc tells us something about the uncertainty of the observations and P tells us something about the uncertainty of the a-prior (first-guess or background) wet refractivity? With this you are finished with section 2 and proceed with your section 3.

Author's response: Thank you for your detailed suggestions. We agree that a restructure of the first two section can help to improve the understanding of our methods. In consequence, we will replace Eq. 10 by Eq. 2 and will derive the relations between STD, refractivity and bending from this equation in the way suggested by the reviewer. Since the focus of this paper lays on atmospheric bending effects, thereby we will not go into detail on how the SWD is obtained from GNSS signals but will show how and to what extend atmospheric bending is compensated in GNSS signal processing using the concept of mapping function - in particular for VMF1. Instead of section '2.2 WV tomography' the manuscript will be extended by a new section '3 The principles of GNSS

tomography'. This will include the basic equations of tomography, presented in a consistent way to section '2 Atmospheric bending effects in GNSS signal processing'. In addition, we will add a more detailed explanation of the singular value decomposition and weighting method applied, as requested by reviewer 2.

Specific comment (3): I suggest that somewhere in the manuscript you plot the following difference dSWD = SWD_T - SWD_0 as a function of the elevation angle for some station. Here SWD_0 is the SWD calculated along the straight line path and SWD_T is the SWD calculated along the ray-path. In essence, dSWD = sum_i 10**(-6) Nw_i ds_i - sum_i 10**(-6) Nw_i dg_i I guess you will find that the following inequality holds true for any elevation angle SWD_0 > SWD_T due to the fact that the ray-path traverses the troposphere at higher altitudes than the straight line path. This would imply that when ray-path bending is not taken into account in tomography the reconstructed troposphere is too dry. To see this you could chose some 'true' wet refractivity field, say N_w0, and simply replace in your eq 7 the term (SWD - A*N_w0) by dSWD. I strongly recommend to do this somewhere in the manuscript.

Author's response: In Figure 6 of the manuscript we plotted the additional ray paths caused by straight-line assumption and discussed its impact on the tomography results. As highlighted by the reviewer, it will lead to a drying effect in the reconstructed refractivity field. To provide some numbers, we computed the differences in SWD (dSWD = SWD_T – SWD_0) and the corresponding paths lengths within the voxel model as suggested by the reviewer for the 6 GNSS sites mentioned in Table 1. Therefore, we made use of the ALARO model data as 'true refractivity field' and ray-traced all lines-of-sight to the GNSS satellites in view, in total 720 observations distributed over 8 epochs (0, 3, 6 . . . UTC) on DoY 121, 2013. The resulting difference (dSWD) are always positive and up to 2.2 cm for ~5 degrees elevation angle. In addition, we computed the resulting drying effect in the refractivity field, which follows from the additional ray path: dN1 = SWD_T[mm] * (ds_0 [km] - ds_T [km]) In addition, we computed the drying effect, caused by the fact, that the straight-line traverses the troposphere at lower altitudes

(assuming exponential decrease of Nw with height between the model layers) dN2 = (SWD_T[mm] – SWD_0[mm]) * ds_0 [km] Both effects have to be taking into account when computing the drying effect in the reconstructed refractivity field obtained along the signal path if bending is not considered. Author's changes in manuscript: In the revised version of the manuscript we will plot the additional paths lengths (Figure 6a) together with dN1 and dN2 as function of elevation angle (as Figure 6b and 6c) and will add a short description in the text how it was computed.

Technical corrections: Abstract L7: '...Thereby, the ray-tracing approach itself but primarily the quality of the a prior field has a significant impact on the reconstruction quality...' improve the writing.

Author's response: This part of the Abstract will be rewritten as follows: 'Thereby, not only the ray-tracing method but also the quality of the a priori field has a significant impact on the quality of the tomography retrievals.'

Introduction L15: 'GNSS' abbreviation not introduced here.

Author's response: Abbreviation 'GNSS' will be spell out the first time when it is used in the manuscript as 'Global Navigation Satellite Systems'

Section 3.1 L18: The inner loop you use is to solve the so called 'homing in problem' (you make use of a shooting method). Please state this more clearly here.

Author's response: In the revised manuscript, we will modify section 3.1 L16 ff as follows: 'After setting the initial parameters, the 'true' ray path is reconstructed iteratively by making use of ray-tracing shooting techniques. Therefore, total refractivity derived from an a priori field is read in and pre-processed for ray-tracing. Hereby, the input data is interpolated vertically and horizontally to the vertical plane, spanned by the y- and z-axis. In the ray-tracing loop, for each height layer $ h_{i+1} $ with $ i=1:(t-1) $ where $ t $ defines the top layer of the voxel model, the geocentric coordinates and the corresponding angles are computed as follows: [..]

Section 3.1 L24: What is the 'outgoing' elevation angle? Please provide a clear definition here.

Author's response: The outgoing elevation angle is determined through the straight-line geometry between the satellite and entry-point of the signal into the atmosphere. In the revised version of the manuscript, we will call it 'the vacuum elevation angle' with reference to Figure 1.

Section 3.2 L4: '...In consequence, the reconstructed...'. The phrase 'In consequence' can be avoided here and at various other places.

Author's response: The sentence 'In consequence, the reconstructed signal travels significantly apart from the observed signal.' will be removed and the entire manuscript will be revised accordingly.

Section 3.2.2: I suggest to show in Fig 4. directly the difference in SWD[m] and not the difference in the bending angle [arcsec]. Also, I do not find the formula for the bending angle in the manuscript. I guess you mean something like arcos(v1,v2) where v1 is the tangent unit vector of the ray-path at the satellite and v2 is the tangent unit vector of the ray-path at the station?

Author's response: The bending angle error in Figure 4 is the difference in elevation angle between full ray-tracing (until ~80 km altitude) and ray-tracing to the upper rim of the tomography model + bending model. Due to the distance of the satellite, it is widely consistent with the bending angle error obtained by analysis of the tangent unit vectors. However, we agree that errors in bending angle are difficult to interpret. Thus, we will convert it into ray path errors and displacement, i.e. how much the ray entry-point differs due to errors in the bending model. We assume that this parameter is more of interest than SWD, since we would like to focus in this paper more on geometrical aspects and its impact on the tomography solution.

Section 4.1: I suggest to add in Fig 8. the difference for the a-prior (first guess or

background) refractivity (ALARO). Is the radiosonde data assimilated into ALARO?

Author's response: This radiosonde data is not assimilated into ALARO. We will add the differences between radiosonde and ALARO to Figure 8 in the revised version of the manuscript.

References: Check all references carefully. For example, Fritsche, M., Dietrich, R., Knofel, C., Rulke, A., and Vey, S.: Impact of higher-order ionospheric terms on GPS estimates, Geophys. Res. Lett., 32, 1–5, 2005. Bender, M., Stosius, R., Zus, F., Dick, G., Wickert, J., Raabe, A. (2011): GNSS water vapour tomography – Expected improvements by combining GPS, GLONASS and Galileo observations. - Advances in Space Research, 47, 5, pp. 886âËŸAËĞ T897. DOI:http://doi.org/10.1016/j.asr.2010.09.011 In the manuscript the correct citation should be e.g. Böhm et al 2006 and not Böhm et al 2006a. Likewise the correct citation should be Hobiger et al 2008 and not Hobiger et al 2008a (there is no b).

Author's response: Thank you. The errors in the references will be corrected and carefully checked again before submission of the revised manuscript.

Additional Reference: Zus, F., Dick, G., Heise, S. and Wickert, J.: A forward operator and its adjoint for GPS slant total delays, Radio Science, 50, 393– 405, doi: 10.1002/2014RS005584, 2015.'

Author's response: Will be added to the revised version of the manuscript